# Enantioselective formal (3 + 3) cycloaddition of bicyclobutanes with nitrones enabled by asymmetric Lewis acid catalysis

Wen-Biao Wu [1,2,3,5], Bing Xu[4,5], Xue-Chun Yang[1,5], Feng Wu[1], Heng-Xian He[1], Xu Zhang[2] & Jian-Jun Feng [1] ✉

The absence of catalytic asymmetric methods for synthesizing chiral (hetero) bicyclo[n.1.1]alkanes has hindered their application in new drug discovery. Here we demonstrate the achievability of an asymmetric polar cycloaddition of bicyclo[1.1.0]butane using a chiral Lewis acid catalyst and a bidentate chelating bicyclo[1.1.0]butane substrate, as exemplified by the current enantioselective formal (3 + 3) cycloaddition of bicyclo[1.1.0]butanes with nitrones. In addition to the diverse bicyclo[1.1.0]butanes incorporating an acyl imidazole group or an acyl pyrazole moiety, a wide array of nitrones are compatible with this Lewis acid catalysis, successfully assembling two congested quaternary carbon centers and a chiral aza-trisubstituted carbon center in the pharmaceutically important hetero-bicyclo[3.1.1]heptane product with up to 99% yield and >99% ee.

The development of asymmetric catalytic reactions, as well as the introduction of new catalytic methods and strategies for preparing chiral molecules in enantioenriched forms, is crucial for drug discovery and innovation. This importance is underscored by the fact that over half of all pharmaceuticals currently in use are chiral compounds containing carbon stereocenters[1].

Recently, the substitution of planar aromatic rings with bridged bicyclic scaffolds has been increasingly acknowledged as a potent approach to enhance the physicochemical and pharmacokinetic properties of drug analogs[2-19]. 1,3-Disubstituted phenyl rings and N-heterocycles are ubiquitous structural motifs found in a variety of small-molecule drugs, with pyridines ranking as the second most prevalent in marketed drugs (Fig. 1a)[20]. In this context, Anderson[21] and Uchiyama[22] independently demonstrated that bridgehead-substituted bicyclo[3.1.1]heptanes (BCHeps), prepared via ring-opening reactions of [3.1.1]propellanes, are suitable bioisosteres for *meta*-substituted benzenes (Fig. 1b). Subsequently, Mykhailiuk documented that aza-BCHeps exhibit remarkable potential as pyridine bioisosteres[23]. Besides noncatalytic linear synthetic methods, several state-of-the-art

catalytic strategies, including photocatalysis, boronyl radical catalysis, Lewis acid catalysis, and silver-promoted methods, have been developed by research groups led by Stephenson[24], Molander[25], Li[26], Waser[27], Wang[28], Deng[29], and Glorius[30] for synthesizing BCHeps and aza-BCHeps through cycloadditions of bicyclo[1.1.1]pentanes (BCPs) or bicyclo[1.1.0]butanes (BCBs) (Fig. 1c)[31-35].

In 2021, Baran and co-workers successfully achieved enantiopure BCP bioisosteres through SFC separation. Furthermore, their study revealed that substituting benzenoids with either *R*- or *S*-enantiomers of BCPs results in distinct drug bioactivities[36]. Consequently, the development of catalytic asymmetric synthesis techniques for (hetero) bicyclo[n.1.1]alkanes is not only highly desirable but would also significantly accelerate the creation of drugs with superior properties. While chirality transfer and bio-catalysis strategies have been used to synthesize chiral BCPs and BCHs respectively[37,38], before the elegant asymmetric (3 + 2) cycloadditions of BCBs reported by Bach[39] and Jiang[40], the only asymmetric difunctionalization reactions within the BCB framework were limited to ring opening processes[41,42]. Notably, the strategies employed by Bach[39] and Jiang[40], focus exclusively on

[1]State Key Laboratory of Chemo/Biosensing and Chemometrics, Advanced Catalytic Engineering Research Center of the Ministry of Education, College of Chemistry and Chemical Engineering, Hunan University, Changsha, P. R. China. [2]School of Chemistry & Chemical Engineering, Yangzhou University, Yangzhou, P. R. China. [3]School of Physics and Chemistry, Hunan First Normal University, Changsha, P. R. China. [4]Department of Chemistry, Fudan University, Shanghai, P.R. China. [5]These authors contributed equally: Wen-Biao Wu, Bing Xu, Xue-Chun Yang. ✉e-mail: jianjunfeng@hnu.edu.cn

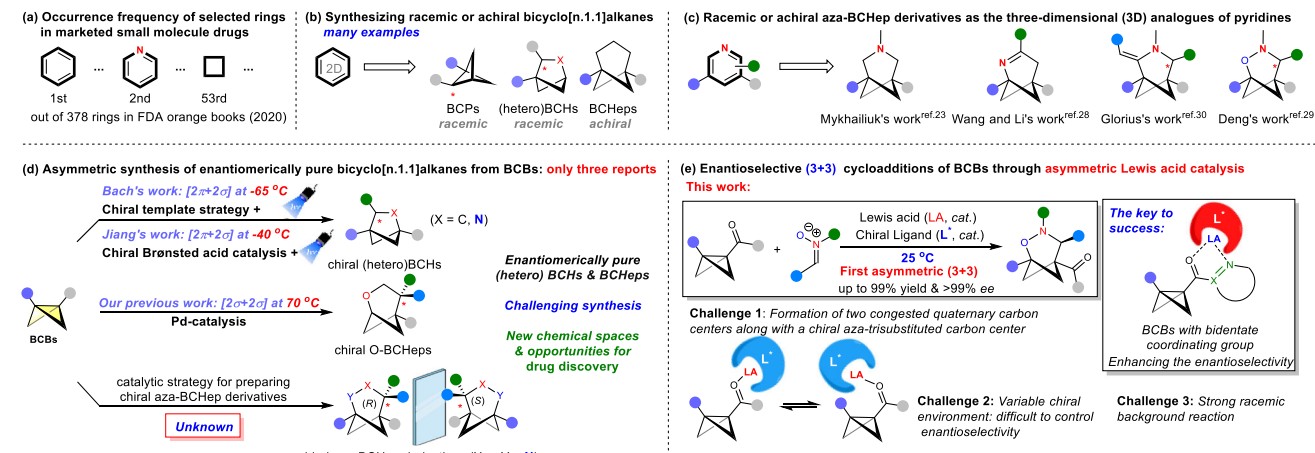

**Fig. 1 | Outline of this work. a** Occurrence frequency of selected rings in marketed small molecule drugs. **b** Synthesizing racemic or achiral bicyclo[n.1.1]alkanes. **c** Racemic or achiral aza-BCHep derivatives as the three-dimensional (3D) analogs of pyridines. **d** Asymmetric synthesis of enantiomerically pure bicyclo[n.1.1]alkanes from BCBs. **e** Enantioselective (3 + 3) cycloadditions of BCBs through asymmetric Lewis acid catalysis.

asymmetric photochemical cycloadditions of BCBs. Our group has recently achieved palladium-catalyzed enantioselective [2σ + 2σ] cycloadditions of BCBs with vinyl oxiranes to synthesize O-BCHeps (Fig. 1d)[43].

Besides radical cycloadditions, polar cycloadditions of BCBs have been developed after Leitch's pioneering research in Lewis acid catalysis[44]. Currently, the Lewis acid catalyzed (3+n) polar cycloadditions of BCBs can utilize ketenes[45], aldehydes[46], heteroarenes[47,48], dienol ethers[49], ynamides[50] and imidazolidines[51] as the cycloaddition partners. During the preparation of this manuscript, Deng pioneered the Eu(OTf)$_3$-catalyzed cycloadditions of BCBs with nitrones[29]. This innovation resulted in the creation of a range of structurally unique and biologically intriguing 2-oxa-3-aza-BCHep skeletons, which show promise as bioisosteres for pyridines. Despite significant progress in this area, the enantioselective synthesis of (hetero)bicyclo[n.1.1]alkanes through Lewis acid catalysis is still unexplored and poses a significant challenge. A major challenge in achieving the asymmetric Lewis acid-catalyzed (3 + 3) cycloadditions of BCBs is identifying optimal ligand capable of overcoming the significant racemic background reaction[29,52], thus facilitating the formation of two congested carbon centers together with a chiral carbon center with high enantioselective control[53,54]. In contrast to donor-acceptor cyclopropanes[55–63], which feature two chelating, electron-withdrawing groups to form a stable chiral catalyst-substrate complex, achieving catalytic asymmetric reactions of BCBs with only a single electron-withdrawing group, leading to chiral environments with variable conformation, poses greater challenges. To address these challenges, we investigated the utilization of BCBs with bidentate coordinating groups and a chiral Lewis acid catalysis strategy in the asymmetric (3 + 3) cycloaddition of BCBs (Fig. 1e).

## Results

### Reaction optimization

Initially, BCB **1a** and nitrone **2a** were chosen as model substrates for reaction optimization (Table 1). Unfortunately, the model reaction showed low enantioselectivity (**3aa**, <40% *ee*) when subjected to zinc-Lewis acid/**L1** or **L2**, as well as privileged chiral oxazoline ligands, including chiral bisoxazolines (BOX) and Py-BOX ligands (Supplementary Figs. 1,2 and Supplementary Table 1). The axially chiral binaphthyl-BOX **L4** yielded promising results. Substituting **2a** with benzyl-substituted nitrone **2b** resulted in the desired **3ab** with a 92% yield and 54% ee (entry 5). Since **L4** exhibited better enantiomeric excess compared to the chiral

spiro-ligand **L3** (entry 3 versus 4), a series of bis(oxazolyl) binaphthyl ligands (($R_a$,R,R)-**L5**-**L7**) were further examined, but no enhancement in enantioselectivity was observed. ($R_a$,S,S)-**L5** was also examined, but it exhibited lower yield and ee compared to ($R_a$,R,R)-**L5** (entry 9 versus 6). Recently, Xie and Guo developed two new tridentate nitrogen ligands, namely PyBPI and PyIPI ligands[64–67]. These ligands exhibited high levels of activity and enantioselectivity in Lewis acid-catalyzed asymmetric reactions. The enantioselectivity of the current reaction with PyBPI **L8** was poor. However, the reaction using PyIPI **L9** yielded **3ab** in 87% *ee* (entry 10 versus 11). Next, we screened an array of commonly used metal-Lewis acids, including Ga(OTf)$_3$, Eu(OTf)$_3$, Cu(OTf)$_2$, Ni(OTf)$_2$ and Co(OTf)$_2$ (entries 12-16). In contrast to other Lewis catalysts, Co(OTf)$_2$ emerged as the optimal catalyst, enhancing enantioselectivity to 93% *ee* (entry 16). It is noteworthy that Ga(OTf)$_3$[44] and Eu(OTf)$_3$[29] previously employed in (3+n) cycloadditions of BCBs, exhibited significant racemic background reactions. Other solvents were also investigated, and CH$_2$Cl$_2$ was found to be superior. Next, PyIPI **L10** − **L12**, which contain more sterically hindered amide substituents, were assessed (entries 17-20). All of these ligands further enhanced the enantioselectivity; The catalytic system comprising Co(OTf)$_2$, **L10**, or **L12** provides **3ab** with a yield of over 90% and an 99% *ee*. Additionally, the catalyst loading could be reduced to 5 mol% almost without deterioration in yield and enantioselectivity (entry 18).

### Substrate scope

With the optimal conditions in hand, we next investigated the reactions of nitrones **2** bearing various R[1] and R[2] motifs with **1a** in the presence of Co(OTf)$_2$/**L10** (Fig. 2). Nitrones with aliphatic R[1] groups, which exhibited low reactivity with Deng's racemic Eu(OTf)$_3$ catalytic system[29], yielded the corresponding enantiomerically pure cycloadducts with excellent yield (80–86%) and enantioselectivity (**3aa**-**ac**). The more reactive *N*-phenyl nitrone **2d** resulted in a lower ee value (88% *ee*) compared to *N*-alkyl nitrones, possibly due to its stronger background reactions. Gratifyingly, the excellent *ee* of **3ad** was reinstated by employing Co(OTf)$_2$/**L12** as the catalyst. The electronic properties of the α-aryl group on the nitrone had a slight impact on the enantioselectivity of the reaction (**3ae**-**au**). Asymmetric (3 + 3) cycloaddition with both electron-rich (methoxy group as in **3ae**, **3ag** and **3al**) and electron-deficient (halides as in **3aj**-**ak** and **3ao**-**ar**) α-aryl nitrones showed outstanding enantioselectivity (97 ~ >99% *ee*). In

## Table 1 | Optimization of the reaction conditions[a]

| Entry | Lewis acid | Ligand | Yield (%)[c] | ee[d] |
|---|---|---|---|---|
| 1 | Zn(ClO$_4$)$_2$·H$_2$O | L1 | 83 | 18[b,e] |
| 2 | Zn(ClO$_4$)$_2$·H$_2$O | L2 | 71 | 33[b] |
| 3 | Zn(ClO$_4$)$_2$·H$_2$O | L3 | 86 | 11[b] |
| 4 | Zn(ClO$_4$)$_2$·H$_2$O | L4 | 70 | 43[b,e] |
| 5 | Zn(ClO$_4$)$_2$·H$_2$O | L4 | 92 | 54[f,e] |
| 6 | Zn(ClO$_4$)$_2$·H$_2$O | ($R_a$,$R$,$R$)-L5 | 99 | 53[f,e] |
| 7 | Zn(ClO$_4$)$_2$·H$_2$O | ($R_a$,$R$,$R$)-L6 | 99 | 27[f,e] |
| 8 | Zn(ClO$_4$)$_2$·H$_2$O | ($R_a$,$R$,$R$)-L7 | 89 | 0[f] |
| 9 | Zn(ClO$_4$)$_2$·H$_2$O | ($R_a$,$S$,$S$)-L5 | 85 | 30[f,e] |
| 10 | Zn(ClO$_4$)$_2$·H$_2$O | L8 | 91 | 27[f] |
| 11 | Zn(ClO$_4$)$_2$·H$_2$O | L9 | 65 | 87[f] |
| 12 | Ga(OTf)$_3$ | L9 | 99 | 0[f] |
| 13 | Eu(OTf)$_3$ | L9 | 90 | 2[f] |
| 14 | Cu(OTf)$_2$ | L9 | 89 | 65[f] |
| 15 | Ni(OTf)$_2$ | L9 | 95 | 80[f] |
| 16 | Co(OTf)$_2$ | L9 | 95 | 93[f] |
| 17 | Co(OTf)$_2$ | L10 | 98 | 99[f] |
| 18 | Co(OTf)$_2$ | L10 | 95 | 98[f,g] |
| 19 | Co(OTf)$_2$ | L11 | 99 | 97[f] |
| 20 | Co(OTf)$_2$ | L12 | 93 | 99[f] |

[a]**1a** (0.1 mmol), **2a** or **2b** (0.12 mmol), Lewis acid (10 mol%) and ligand (12 mol%) in CH$_2$Cl$_2$ at room temperature for 24 h. [b]**2a** was used. [c]NMR yield with CH$_2$Br$_2$ as an internal standard. [d]The enantiomeric excess (ee) of the product was determined by HPLC using a chiral stationary phase. [e]The other enantiomer was obtained. [f]**2b** was used. [g]Co(OTf)$_2$ (5 mol%) and **L10** (6 mol%) were used.

addition to α-aryl nitrones bearing functional groups in the *para*- and *meta*-positions, nitrones **2e**-**f** with *ortho*-substituted phenyl groups also smoothly produced the desired cycloadducts (**3ae-af**) with 99% *ee*. Notably, an alkynyl group was found to be compatible (**3as**), further highlighting the functional-group tolerance of this reaction. Nitrones containing disubstituted aryl groups, such as **3at** (3,4-diOMe) and **3au** (3,5-diMe), or (α- or β-) naphthyl groups (**3av**-**aw**), at the R² position, also yielded desired products in the range of 83–87% with 97–99% *ee*. Furthermore, heteroaryl groups such as 2-indolyl, 2-furyl, and 2-thienyl were well tolerated, resulting in the formation of **3ax, 3ay**, and **3az** with up to

>99% ee. α-Alkyl nitrone **2aa** which had not been compatible with Deng's racemic catalytic system[29], provided **3aaa** in acceptable yield with 70% ee. Significantly, the asymmetric reaction between **1a** and α-alkenyl nitrone **2bb** exhibited remarkable enantioselectivity (97% ee).

After the investigation of the nitrone scope, we studied the scope with respect to the BCBs (Fig. 3). The reaction tolerated alkyl and phenyl substituents at the phenyl moiety well (**3bb-3hb**). An array of BCBs bearing either electron-donating or electron-withdrawing groups on the *para*- (*p*-Me, *p*-Br, *p*-F, *p*-CF₃, **3bb-eb**), and *meta*-positions (*m*-Me, *m*-Cl, **3fb-gb**) of phenyl rings (R¹ group in BCB),

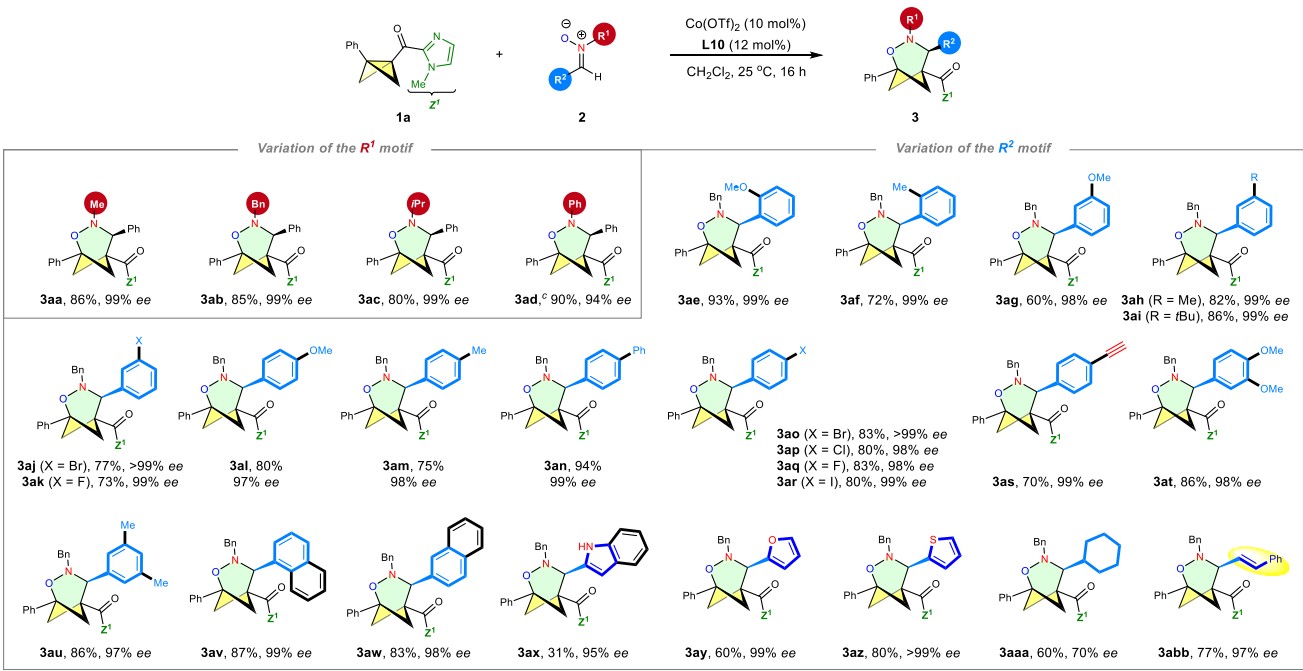

**Fig. 2 | Scope I: variation of the nitrones.** [a,b] [a]Reaction conditions: 0.2 mmol **1a**, 1.2 equiv. **2**, and 10 mol% Co(OTf)$_2$/12 mol% **L10** in 2.0 mL CH$_2$Cl$_2$ at 25 °C for 16 h. [b]Isolated yield. [c]**L12** was utilized in place of **L10**.

participated smoothly in the reaction (84–95% yields, 98–99% *ee*). Additionally, BCB with *o*-tolyl group (**3hb**) and naphthyl-substituted BCB (**3ib**) also efficiently produced the desired cycloadducts with excellent enantioselectivity. Notably, the BCB **1e**, containing a highly electron-withdrawing trifluoromethyl group, was previously considered incompatible with Glorius's Lewis acid catalytic system[46]. However, it still yielded the corresponding cycloadducts (**3eb** versus **3bb**) in good yield and selectivity. These results suggest that it follows the same pathway as Deng's work, involving a concerted nucleophilic ring-opening mechanism of BCBs with nitrones[29]. Subsequently, we investigated the impact of *N*-substitution of 2-acyl imidazole in the current cycloadditions. The identity of the *N*-substituent (**3jb-mb**) had a minor effect on enantioselectivity (98–99% *ee*), but did influence the yield (71–85% yield). Moreover, we found that the Glorius's BCB substrates with acyl pyrazole substituent instead of acyl imidazole could also be well accepted in the enantioselective cycloaddition with **2b**, and the corresponding (3 + 3) cycloadducts were produced with up to 99% yield and >99% *ee*. However, in some cases, lower *ee* values were observed when using Glorius's BCB compared to BCBs featuring a 2-acyl imidazole moiety (**3ob** versus **3eb**). For instance, the (3 + 3) reaction of **2n** with *N*-phenyl nitrone **2d** yields the desired **3nd** with 28% *ee* (Supplementary Fig. 5). In contrast, the cycloaddition of **1a** and **2d** produces **3ad** with 94% *ee* under identical reaction conditions. Unfortunately, the employment of the methyl-substituted BCB delivered the corresponding product with low *ee* (52% *ee*, Supplementary Fig. 5).

## Synthetic applications
To demonstrate the synthetic utility of these chiral cycloadducts, several transformations of the cycloadducts were investigated (Fig. 4). The cycloaddition of **1n** and **2b** is scalable to 1.0 mmol under standard reaction conditions, yielding **3nb** with 99% yield and 99% *ee*. The acyl pyrazole moiety of **3nb** and **3qb** was converted to an aldehyde group using LiAlH$_4$, preserving its chirality (Fig. 4a). Moreover, in a two-step process, we integrated the chiral 2-oxa-3-azaBCHep **3qb** unit into the structures of the *anti*-dyskinesia agent Sarizotan[68] and dopamine D$_2$ receptor

antagonist Adoprazine[69] by substituting the pyridine ring. These instances showcased the practical synthetic value of the chiral (3 + 3) cycloadducts. Cleavage of the imidazole moiety gave rise to the desired ketone **5** in 75% yield. The nucleophilic substitution or addition of **3nb** smoothly produce enantiomerically enriched compounds **6** and **7**, respectively. Cleaving the *N-O* bond in **3nb** and deprotecting the benzyl group in the presence of H$_2$ and Pd(OH)$_2$/C produces functionalized cyclobutane **8** (Fig. 4b). Notably, including a terminal-alkyne group in cycloadduct **3as** facilitates the incorporation of the chiral hetero-BCHep scaffold into bioactive compounds like Adapalene and Isoxepac through click reaction (Fig. 4c).

## Mechanistic studies
In order to gain insight into this transformation, a study was conducted to examine the correlation between the *ee* value of PyIPI-**L10** and that of (*S*)-**3ab** (Fig. 5a). The study revealed a linear relationship, suggesting that an active catalyst/ligand being of a monomeric nature. Based on Deng[29] and Xie&Guo's work[66,67], to unravel the origin of enantiocontrol, density functional theory (DFT) calculations were performed at PBE0/6-31 G(d)-SDD level of theory, using **1a** and **2b** as the model substrates along with the Co(II) − **L12** chiral system. As shown in Fig. 5b, transition states **Ts-S** and **Ts-R** leading to both products were located, in which divalent Co coordinated with two nitrogen and one oxygen atoms of **L12** as well as one oxygen and nitrogen atoms of **1a**, generating a square pyramidal geometry. The difference between the two activation barriers of the intramolecular nucleophilic cyclization for the two enantiomers was 2.2 kcal/mol, consistent with the excellent enantioselectivity experimentally observed. Moreover, noncovalent interactions between reactant fragment and **L12** in **Ts-S** and **Ts-R** were explored using independent gradient model based on Hirshfeld partition (IGMH) analysis[70]. Three pairs of C−H•••O interaction existed in both **Ts-S** and **Ts-R**. Notably, in the favored transition state **Ts-S**, imidazolidone of **L12** engaged in C−H...π interaction with phenyl group of nitrone **2a**, likely dominating the preference for the nucleophilic attack of the enolate

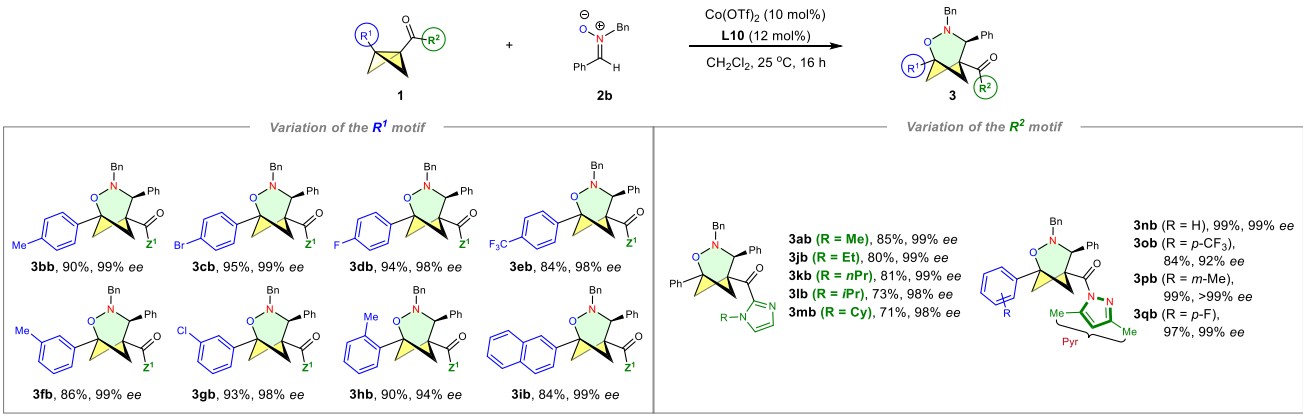

**Fig. 3 | Scope II: variation of the BCBs.** [a,b] [a]Reaction conditions: 0.2 mmol **1**, 1.2 equiv. **2b**, and 10 mol% Co(OTf)$_2$/12 mol% **L10** in 2.0 mL CH$_2$Cl$_2$ at 25 °C for 16 h. [b]Isolated yield.

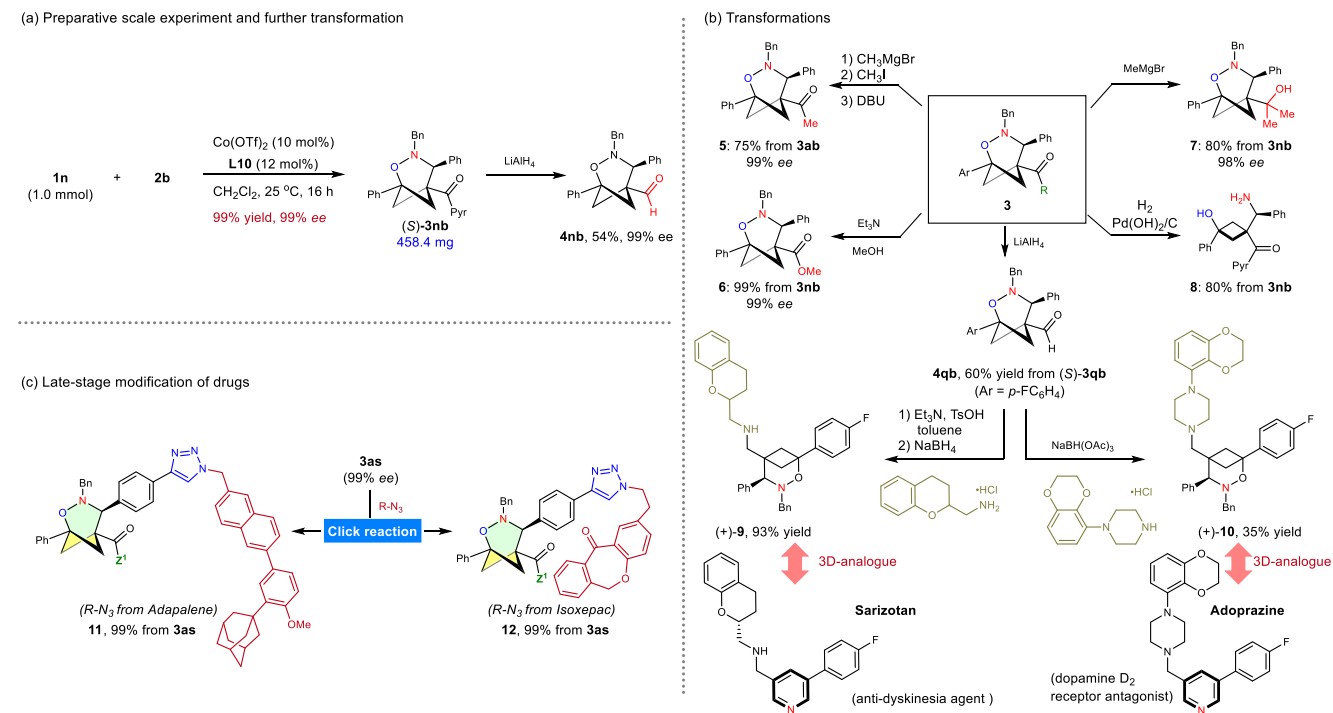

**Fig. 4 | Scale-up and derivatizations. a** Scale-up synthesis of **3nb**. **b** Derivatization of the cycloadducts. **c** Late-stage modification of drugs.

on the *Re*-face of nitrone **2a** to furnish the (*S*)-configuration of product **3ab**.

## Discussion

In conclusion, a strategy utilizing Lewis acid catalysis has been developed for the atom-economic and enantioselective synthesis of bridged bicyclic scaffolds from BCBs. By utilizing a chiral Co(II)/PyIPI catalyst and bidentate chelating BCB substrates, we can efficiently access enantioenriched pharmaceutically important hetero-bicyclo[3.1.1] heptane derivatives by varying the BCBs or nitrones used in the (3 + 3) cycloadditions. The acyl imidazole group or acyl pyrazole moiety in BCBs plays a crucial role in stereocontrol. The synthetic utility of this protocol has been further demonstrated in the concise synthesis of the analog of bioactive pyridine and the late-stage functionalization of drugs. Given the significance of chiral (hetero)bicyclo[n.1.1]alkane

scaffolds as bioisosteres and the need for a new strategy for the asymmetric cycloaddition of BCBs, we anticipate that this approach will yield positive outcomes in both synthetic and medicinal chemistry.

## Methods

### General Procedure for the Enantioselective (3 + 3) Cycloadditions

Under an atmosphere of N$_2$, to a 25 mL oven-dried Schlenk tube were added Co(OTf)$_2$ (7.1 mg, 0.020 mmol) and **L10** (17.1 mg, 0.024 mmol), followed by 2.0 mL of anhydrous CH$_2$Cl$_2$. The solution was stirred at 25 °C for 0.5 h, and then the BCBs **1** (0.20 mmol, 1.0 equiv) and nitrones **2** (0.24 mmol, 1.2 equiv) were added. Then the resulting mixture was stirred at room temperature for 16 h till full conversion of **1** by TLC analysis. After the solvent was removed under reduced pressure, the residue was directly subjected to a column chromatography

**(a) Non-linear effect study**

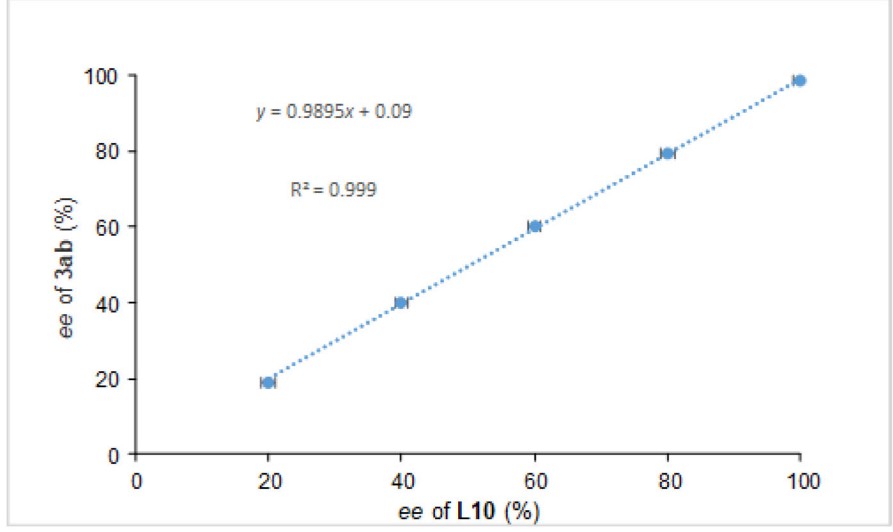

**(b) DFT calculations**

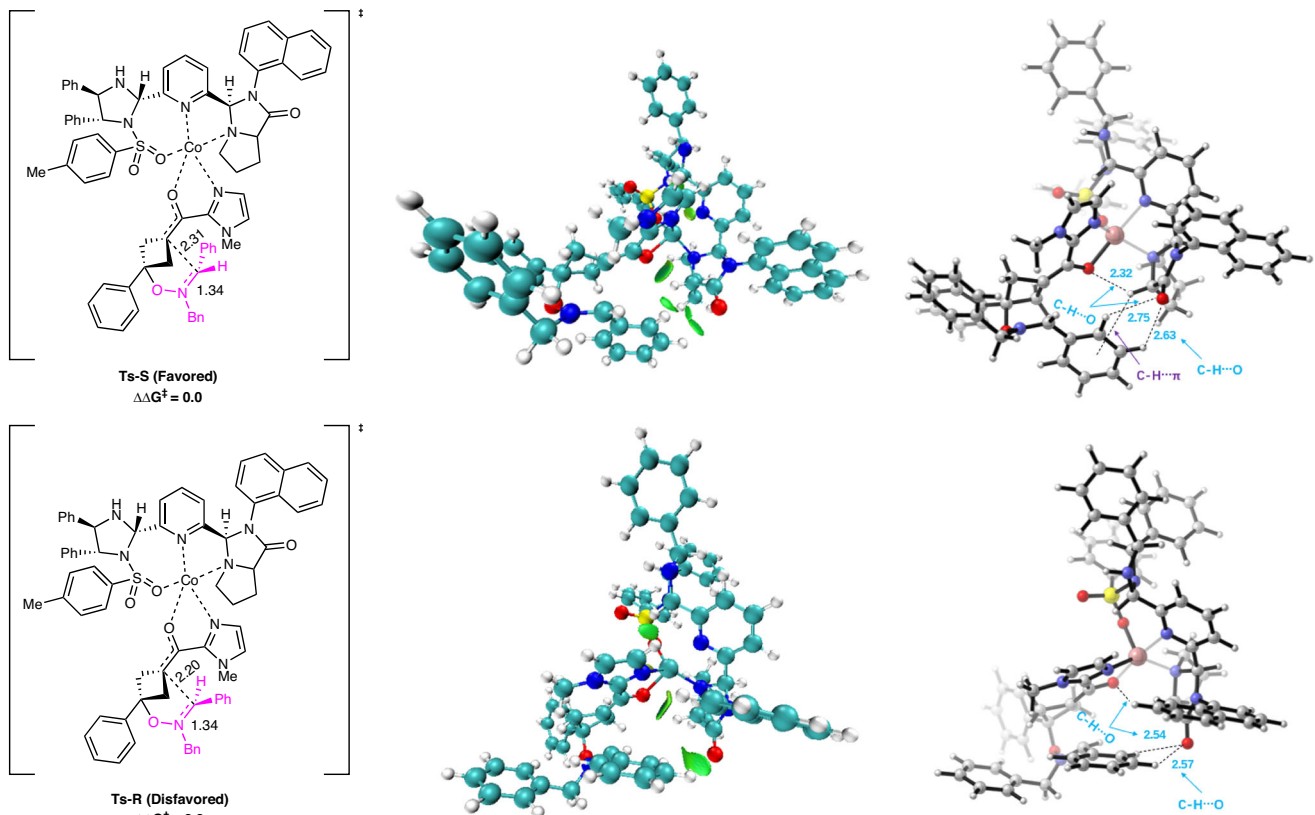

**Fig. 5 | Non-linear effect study and DFT calculations. a** The relationship between *ee* values of ligand **L10** and product **3ab**. **b** Optimized structures of the enantio-determining transition state and IGMH analysis of the non-covalent interactions in **Ts-S** and **Ts-R**. The Gaussian 09 level of PBE0/6-31 G(d)-SDD was used. All distances are in angstrom. The relative free energies are in kcal mol⁻¹.

purification using PE/EtOAc (4:1, v/v) as the eluent, to afford the desired product **3**.

## Data availability

The data supporting the findings of this study are available within this article and its Supplementary Information, which contains experimental details, characterization data, copies of NMR spectra and HPLC spectra for all new compounds, X-ray structural analysis, and DFT calculation data. Crystallographic data for the structures reported in this article have been deposited at the Cambridge Crystallographic Data Centre, under deposition numbers CCDC 2345666 ((*S*)-3ao). Copies of the data can be obtained free of charge via https://www.ccdc.cam.ac.uk/structures/. Source data of DFT calculation are also provided with this paper. All other data are available from the corresponding author upon request. Source data are provided with this paper.

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

## Acknowledgements

We are grateful to the Fundamental Research Funds for the Central Universities, the National Natural Science Foundation of China (22471068 for J.-J. F), Natural Science Foundation of Hunan Province (2024JJ6126 for W.-B.W.) and the China Postdoctoral Science Founda-tion (2022M713667 for B.X., 2024M750865 for W.-B.W.) for financial support. ¹H, ¹³C NMR spectra, single crystal X-ray diffraction and HRMS were performed at Analytical Instrumentation Center of Hunan University.

## Author contributions

J.J.F. conceived the study. W.B.W and X.C.Y. carried out the experi-ments and data analysis work. B.X. performed the computational stu-dies. F.W. and H.X.H. synthesized the bicyclobutanes. X.Z. synthesized the nitrones. The paper was written by J.J.F. All authors contributed to discussions. W.B.W., B.X. and X.C.Y. contributed equally.

## Competing interests

The authors declare no competing interests.
