## [Peer Review File · Nature Communications]

Enantioselective formal (3+3) cycloaddition of bicyclobutanes with nitrones enabled by asymmetric Lewis acid catalysisREVIEWER COMMENTS

Reviewer #1 (Remarks to the Author):

The authors have presented a novel Lewis acid catalyzed asymmetric $[4\pi+2\sigma]$ cycloaddition of BCBs with nitrones, efficiently offering pharmaceutically important heterobicyclo[3.1.1]heptanes with excellent yields and enantioselectivities. Preparative-scale reactions and various product transformations, especially the concise synthesis of the analogue of bioactive pyridine and the late-stage functionalization of drugs, further highlighted the potential synthetic utility of this reaction.

In comparison with Deng's racemic work (Angew. Chem. Int. Ed. 2024, e202318476), a big breakthrough is realizing the asymmetric version by utilizing a chiral Co(II)/PyIPI catalyst and an ingeniously designed bidentate chelating BCB substrates, accompanied with better reaction efficiency and more general substrate scope including N-alkyl nitrones. This reviewer suggests that this work would deserve publication in Nature Communications with some minor suggestions as follows:

- It is better to match the ligand Nos in the main text and SI for the convenience of readers.
- In SI, the source of ligands needs to be addressed. If directly purchased, CAS numbers and suppliers need to be indicated. If synthesized, related literatures should be cited properly, and it would be much better to provide the general synthetic routes and operations for the different types of ligands.
- Page 6, Line 100, a typo error exists for "conditionsa.a"
- The ee value (93) in Figure 2, entry 16 of the main text is not consistent with that (93.3) in Table S1 entry 9 of SI. The authors should also verify the consistency of other data.
- Melting points of solid products should be given if conditional.
- A latest literature on BCB cycloaddition (Org. Lett. 2024, 26, 4104–4110) should be properly cited.

Reviewer #2 (Remarks to the Author):

The authors report an asymmetric polar cycloaddition of BCB by a chiral Lewis acid catalysis. The key novelty worth considering is the utilization of BCBs with bidentate coordination groups and chiral Lewis acid catalysts to achieve asymmetric $[4\pi+2\sigma]$ cycloaddition reactions, enhancing the enantioselectivity. In many ways, the cycloaddition reaction by Lewis acid-catalyzed has been reported previously (ex: Angew. Chem. Int. Ed. 2022, 61, e202211596; Angew. Chem. Int. Ed. 2023, 62, e202308606; J. Am. Chem. Soc. 2023, 145, 22, 12233-12243). Even cycloaddition reactions of bicyclo[1.1.0]butanes (BCBs) with tension release-driven have also been used to synthesize multiple classes of potentially chiral drug intermediates (Angew. Chem. Int. Ed. 2023, 62, e202304771; Angew. Chem. Int. Ed. 2024, e202405781). However, I believe that the utilization of bidentate coordinated BCB with chiral Lewis acid catalysts to obtain heterobicyclo[3.1.1] heptane derivatives with important pharmacological effects is very novel, and this strategy is also inspired in the future synthesis of multiple classes of chiral drugs. As such, considering the potential and positive in synthesis and medicinal chemistry, it can be published in Nature Communications. But, some contents in the manuscript require further refinement.

1. The manuscript mentions that chiral Lewis acid catalysts exert asymmetric selective catalysis, and the effect of ligand chirality was no mentioned. Is the effect of the ligand's own chirality on selective catalysis considered? The different configurations of the ligand affect

the asymmetric reaction?

2. The Lewis acid catalysts with different metal centers were selected in the synthesis and the ligands were subsequently examined and optimized. What are the key factors and reasons for the high activity of the Lewis acid catalysts? And what is the basis for optimizing the selection of the best conditions? For example, the spatial configuration of the catalyst, the Lewis acid strength of the metal center.
3. The key role of Lewis acidic metal centers is confirmed by comparing the catalytic activity of different Lewis metal centers of the same ligand. In addition, is a single ligand catalytic in the absence of a Lewis acidic metal center?
4. Lewis catalysts with L1-L5 as ligands have been reported, whereas L8-L12 was used as a new ligand for the synthesis of metal Lewis acid catalysts. How to determine the coordination forms and structural formulae, as well as confirming that it is the spatial site resistance that leads to the difference in activity.
5. The optimization of solvents, only CH₂Cl₂ is shown, please add in the effect of other solvents on the asymmetric synthesis reaction. For example, toluene, CHCl₃, etc.
6. Reaction temperature affects asymmetric catalytic reactions? Please add optimization experiments of temperature.
7. The activity of the catalyst strongly depends on the drying process, especially its solubility. However, only a few lanthanide salts have some degree of solubility in organic solvents. Therefore, a common strategy is to add a suitable co-solvent in reaction system. What is the state of the Lewis catalyst present in the reaction system in this study? Are co-solvents or additives considered to optimize the catalytic reaction?
8. Nitrones 2 bearing various R₁ and R₂ motifs showed large differences in reactivity and enantioselectivity. What are the reasons for the differences? How to confirm it?
9. More detailed experiments and descriptions of the catalytic mechanism of Lewis-catalyzed asymmetric cycloaddition reactions should be provided.
10. To deeply explore the possible mechanism of the reaction, the authors could utilize isotope labeling experiments and HRMS in order to trace the product in the reaction process.
11. Asymmetric catalytic efficiencies in the presence and absence of metal complexes should be compared to ensure metal Lewis acid catalysis.
12. The identification of key intermediates is an important part of describing the catalytic mechanism. The key intermediates in the reaction process are characterized and experimentally confirmed. DFT calculations can help to elucidate the catalytic mechanism by simulating the reaction process and obtaining its reactive intermediates, transition states, and so on. In the manuscript, a diagram of the catalytic mechanism should be presented.
13. The formation of complexes between BCB and Lewis acid catalysts helps to enhance enantioselectivity. How is the intermediate complex/transition state determined? The intermediate structure can be well speculated using DFT computational simulations.

Reviewer #3 (Remarks to the Author):

In this manuscript, Feng and co-workers described the first example of (highly) enantioselective formal [4+2] cycloaddition of BCBs with nitrones using a chiral Lewis acid catalyst. Addressing the inherent difficulties of such a reaction required a meticulous screening of conditions. Key to success was the combination of a Co(II) salt with an unusual tridentate nitrogen ligand of the PyIPI family. Substrates incorporating imidazole or pyrazole substituents were used in order to ensure the coordination of the latter to the catalyst. A monomeric transition state was proposed based on non-linear effect studies. Consistently, the reaction delivered enantio aza-oxa-bicycloheptanes, which are interesting as saturated

analogs of disubstituted pyridines. These products could be converted into a series of valuable derivatives, including a densely functionalized cyclobutanol. The scope of the reaction was broad: a diversity of substituents on both the BCBs (bridgehead position) and the nitrones (N and C positions) were well tolerated. Enantiomeric excesses as high as 99% and good to excellent yields were obtained in most cases.

Overall, Feng and co-workers present some elegant and solid chemistry. Assessing the novelty delivered by their manuscript was initially less trivial, because of the analogies with the communication published by Deng and co-workers in *Angewandte Chemie* earlier this year (ACIE 2024, 63, e202318476). In fact, the latter described a very similar annulative reaction involving BCBs and nitrones. Upon accurate reflection, I nonetheless got to the conclusion this occurrence does not invalidate the originality or the significance of Feng's work, which are mainly residing in the asymmetric character of the transformation. Enantio-induction in annulations of BCBs is extremely challenging to achieve, especially under Lewis acid catalysis. Providing a well-defined chiral environment through the coordination of the catalyst is though, so that background reactions can easily dominate. So far, the only reports depicting asymmetric annulations of BCBs have been the ones by Bach and, respectively, Jiang. These valuable achievements, which relied on photocatalysis, were however limited to formal [2+2] cycloadditions and subjected to structural constraints of the reacting species. The robust and general method established by Feng and co-workers indicates that a broader use of bicyclobutanes in enantioselective annulation is feasible, which is highly desirable when investigating the resulting cycloadducts for drug discovery purposes. Based on these considerations, I therefore strongly encourage the publication of this manuscript by Feng and co-workers in *Nature Communications*. The authors are however recommended to implement the following revisions:

- It has now become common to read of $[n\pi+2\sigma]$ in reports describing annulations with bicyclobutanes. Calling the electrons coming from the latter as 2σ is partially erroneous, because the bond across BCBs has a ca. 95% π character, as computed by Newton and Schulman already in 1972 (JACS 1972, 94, 767). The authors should therefore change $[4\pi+2\sigma]$ into $[4+2]$ everywhere in their manuscript to provide a less ambiguous definition of the process.
- When examining the scope, the variations of the BCB substrates are limited to changing the aryl substituent at the bridgehead position of the bicyclic scaffolds. If not already done, the authors should also tested H- and/or alkyl substituted BCBs and comment about the corresponding results.
- The manuscript is well written. The abstract, the introduction and the conclusions are clear, and offer an adequate context to justify the importance of the reaction. The Supporting Information, on the other side, requires some adjustments. The authors should:
 - Give synthetic protocols for the preparation of all the tested chiral ligands that are non-commercially available, including the related characterization data.
 - Check and correct the significant digits throughout the document, as they are not consistent.
 - Provide the melting points of all the previously unreported compounds that are solid.
 - Concerning the linear effect experiment, error bars should be displayed to prove the statistical significance of the shown results.
- The authors provided an exhaustive list of citations. References 39 and 40 appear however poorly fitting in the context in which they are given. Rephrasing lines 55-56 in page 4 may be

appropriate, by mentioning that – prior to Bach and Jiang – the only asymmetric difunctionalization reactions across the BCB framework were limited to ring opening processes.

Response to Referees

Response to Referee (1)'s Comments -- changes highlighted in yellow.

Comments:

The authors have presented a novel Lewis acid catalyzed asymmetric $[4\pi+2\sigma]$ cycloaddition of BCBs with nitrones, efficiently offering pharmaceutically important heterobicyclo[3.1.1]heptanes with excellent yields and enantioselectivities. Preparative-scale reactions and various product transformations, especially the concise synthesis of the analogue of bioactive pyridine and the late-stage functionalization of drugs, further highlighted the potential synthetic utility of this reaction.

In comparison with Deng's racemic work (Angew. Chem. Int. Ed. 2024, e202318476), a big breakthrough is realizing the asymmetric version by utilizing a chiral Co(II)/PyPI catalyst and an ingeniously designed bidentate chelating BCB substrates, accompanied with better reaction efficiency and more general substrate scope including N-alkyl nitrones. This reviewer suggests that this work would deserve publication in Nature Communications with some minor suggestions as follows:

Response: We appreciate the time and effort you dedicated to providing feedback on our manuscript. Thank you for your positive comments and valuable suggestions to improve the quality of our manuscript. As the principal investigator (PI) of a newly established research group, I am delighted that you, a prominent figure in the field, have acknowledged our work, and your encouragement is highly valued.

1: It is better to match the ligand Nos in the main text and SI for the convenience of readers.

Response 1: Thank you for your valuable suggestion. As per your request, we have revised the ligand numbers in the SI to match those in the main text.

2: In SI, the source of ligands needs to be addressed. If directly purchased, CAS numbers and suppliers need to be indicated. If synthesized, related literatures should be cited properly, and it would be much better to provide the general synthetic routes and operations for the different types of ligands.

Response 2: Thank you for your suggestion. The origins of the ligands discussed in the article are detailed within the SI.

3: Page 6, Line 100, a typo error exists for “conditionsa.a”.

Response 3: Thank you for your suggestion. The typographical error in the main text have been corrected.

4: The ee value (93) in Figure 2, entry 16 of the main text is not consistent with that (93.3) in Table S1 entry 9 of SI. The authors should also verify the consistency of other data.

Response 4: Thank you for your suggestion. We have adjusted the enantiomeric excess from 93.3% to 93%.

5: Melting points of solid products should be given if conditional.

Response 5: Thank you for your suggestion. The melting points of the solid products are listed in the ESI.

6: A latest literature on BCB cycloaddition (Org. Lett. 2024, 26, 4104–4110) should be properly cited.

Response 6: Thank you for your suggestion. The outstanding study on the $B(C_6F_5)_3$ -catalyzed higher-order (5+3) and (6+3) cycloadditions of BCBs has been referenced as ref 51.

Response to Referee (2)'s Comments -- changes highlighted in green.

Comments:

The authors report an asymmetric polar cycloaddition of BCB by a chiral Lewis acid catalysis. The key novelty worth considering is the utilization of BCBs with bidentate coordination groups and chiral Lewis acid catalysts to achieve asymmetric $[4\pi+2\sigma]$ cycloaddition reactions, enhancing the enantioselectivity. In many ways, the cycloaddition reaction by Lewis acid-catalyzed has been reported previously (ex: Angew. Chem. Int. Ed. 2022, 61, e202211596; Angew. Chem. Int. Ed. 2023, 62, e202308606; J. Am. Chem. Soc. 2023, 145, 22, 12233-12243).

Even cycloaddition reactions of bicyclo[1.1.0]butanes (BCBs) with tension release-driven have also been used to synthesize multiple classes of potentially chiral drug intermediates (Angew. Chem. Int. Ed. 2023, 62, e202304771; Angew. Chem. Int. Ed. 2024, e202405781). However, I believe that the utilization of bidentate coordinated BCB with chiral Lewis acid catalysts to obtain heterobicyclo[3.1.1] heptane derivatives with important pharmacological effects is very novel, and this strategy is also inspired in the future synthesis of multiple classes of chiral drugs. As such, considering the potential and positive in synthesis and medicinal chemistry, it can be published in Nature Communications. But, some contents in the manuscript require further refinement.

Response: We appreciate the time and effort you dedicated to providing feedback on our manuscript. Thank you for your positive comments and valuable suggestions to improve the quality of our manuscript. As the principal investigator (PI) of a newly established research group, I am delighted that you, a prominent figure in the field, have acknowledged our work, and your encouragement is highly valued.

1: The manuscript mentions that chiral Lewis acid catalysts exert asymmetric selective catalysis, and the effect of ligand chirality was no mentioned. Is the effect of the ligand's own chirality on selective catalysis considered? The different configurations of the ligand affect the asymmetric reaction?

Response 1: Thank you for your suggestion. As per your request, in Table S1, entry 27, when the enantiomer of **L10** (*ent*-**L10**) was employed as the ligand, the opposite enantiomer of (*S*)-**3ab** was produced.

2: The Lewis acid catalysts with different metal centers were selected in the synthesis and the ligands were subsequently examined and optimized. What are the key factors and reasons for the high activity of the Lewis acid catalysts? And what is the basis for optimizing the selection of the best conditions? For example, the spatial configuration of the catalyst, the Lewis acid strength of the metal center.

Response 2: Thank you for your suggestion. The control experiment demonstrated that the reaction did not proceed without Lewis acid catalysts. In contrast, utilizing $\text{Co}(\text{OTf})_2$ alone resulted in a 99% yield of the desired racemic **3ab**, highlighting the essential role of the Lewis

acid catalyst. Following the Hard Soft Acids Bases (HSAB) principle, we investigated several chiral N-containing ligands, leading to the discovery that the PyIPI ligand **L10**, developed by Xie and Guo's group (ref 64-67), exhibited the highest enantioselectivity. Corresponding information is shown in ESI Table S1, entries 24-27.

3: The key role of Lewis acidic metal centers is confirmed by comparing the catalytic activity of different Lewis metal centers of the same ligand. In addition, is a single ligand catalytic in the absence of a Lewis acidic metal center?

Response 3: Thank you for your suggestion. The control experiment showed that the reaction did not progress in the absence of Lewis acid catalysts, and a single ligand **L10** was unable to catalyze the reaction without $\text{Co}(\text{OTf})_2$. Corresponding information is shown in ESI Table S1, entries 25-26.

4: Lewis catalysts with L1-L5 as ligands have been reported, whereas L8-L12 was used as a new ligand for the synthesis of metal Lewis acid catalysts. How to determine the coordination forms and structural formulae, as well as confirming that it is the spatial site resistance that leads to the difference in activity.

Response 4: Ligand **L8**, namely PyBPI, and Ligands **L9-12**, namely PyIPI, were synthesized by Xie and Guo's group (ref 64-67). Based on their mechanistic studies on the $\text{Co}(\text{OTf})_2$ -PyIPI-catalyzed asymmetric dearomative [3+2] annulation of quinolines and donor-acceptor aminocyclopropanes (ref 67), we also conducted density functional theory (DFT) calculations to elucidate the enantiocontrol mechanism in our asymmetric (3+3) cycloadditions of BCBs and nitrones (Fig 6).

5: The optimization of solvents, only CH_2Cl_2 is shown, please add in the effect of other solvents on the asymmetric synthesis reaction. For example, toluene, CHCl_3 , etc.

Response 5: Thank you for your suggestion. As per your request, other solvents like toluene and CHCl_3 were investigated. Corresponding information is shown in ESI Table S1, entries 11 and 20.

6: Reaction temperature affects asymmetric catalytic reactions? Please add optimization experiments of temperature.

Response 6: Thank you for your suggestion. In response to your request, we included temperature optimization experiments in ESI Table S1, entries 18 and 19. Conducting the reaction at 40°C had almost no effect on the yield and enantioselectivity. In contrast, reaction temperatures at 0°C led to a decrease in yield while maintaining enantiomeric excess (*ee*) values.

7: The activity of the catalyst strongly depends on the drying process, especially its solubility. However, only a few lanthanide salts have some degree of solubility in organic solvents. Therefore, a common strategy is to add a suitable co-solvent in reaction system. What is the state of the Lewis catalyst present in the reaction system in this study? Are co-solvents or additives considered to optimize the catalytic reaction?

Response 7: Thank you for your suggestion. In accordance with your request, co-solvents like HFIP (Hexafluoroisopropanol) and H₂O, along with additives such as 4 ÅMS, were incorporated into the optimization experiments detailed in ESI Table S1, entries 21-23. Stirring Co(OTf)₂ and **L10** in anhydrous CH₂Cl₂ at 25°C for 0.5 h resulted in the formation of a homogeneous system.

8: Nitrones **2** bearing various R¹ and R² motifs showed large differences in reactivity and enantioselectivity. What are the reasons for the differences? How to confirm it?

Response 8: Thank you for your suggestion. Nitrones with diverse R¹ and R² motifs exhibited distinct background reactions. For example, nitrones with aliphatic R¹ groups, which exhibited low reactivity with Deng's racemic Eu(OTf)₃ catalytic system (ref 29), yielded the corresponding enantiomerically pure cycloadducts with excellent yield (80-86%) and enantioselectivity (**3aa-ac**). The more reactive *N*-phenyl nitrone **2d** resulted in a lower *ee* value (88% *ee*) compared to *N*-alkyl nitrones, possibly due to its stronger background reactions. In the absence of a chiral ligand, we observed that Co(OTf)₂ catalyzed the racemic (3+3) cycloaddition between nitrone **2b** with an aliphatic *Bn* group and BCB **1a**, resulting in the desired product *rac*-**3ab** with an 85% NMR yield in 20 min (87% conversion of **1a**); the Co(OTf)₂-catalyzed racemic (3+3) cycloaddition of *N*-phenyl nitrone **2d** with BCB **1a** produced the desired product *rac*-**3ad** with a 99% NMR yield (100% conversion of **1a**) in 20 min. Corresponding information is shown in Scheme S4 in the ESI,

9: More detailed experiments and descriptions of the catalytic mechanism of Lewis-

catalyzed asymmetric cycloaddition reactions should be provided.

Response 9: Thank you for your suggestion. Building upon Deng's mechanistic studies of Lewis-catalyzed racemic (3+3) cycloadditions involving BCBs and nitrones (ref 29), and our density functional theory (DFT) calculations to elucidate the enantiocontrol mechanism, a detailed description of the proposed catalytic cycle and computed transition structures (TSs) for the present asymmetric (3+3) cycloaddition reactions is presented in the ESI section 10, Page 17.

10: To deeply explore the possible mechanism of the reaction, the authors could utilize isotope labeling experiments and HRMS in order to trace the product in the reaction process.

Response 10: Thank you for your suggestion. Deng recently conducted DFT mechanistic studies on the Lewis-catalyzed racemic (3+3) cycloadditions involving BCBs and nitrones (ref 29). Additionally, Xie and Guo's group also performed DFT mechanistic studies on the Co(OTf)₂-PyIPI **L12**-catalyzed asymmetric dearomative [3+2] annulation of quinolines and donor-acceptor aminocyclopropanes (ref 67). Building upon their comprehensive mechanistic investigations, we did not conduct isotope labeling and HRMS experiments. Instead, we utilized density functional theory (DFT) calculations to elucidate the enantiocontrol mechanism in our asymmetric (3+3) cycloadditions of BCBs and nitrones (Fig 6).

11: Asymmetric catalytic efficiencies in the presence and absence of metal complexes should be compared to ensure metal Lewis acid catalysis.

Response 11: Thank you for your suggestion. The control experiment demonstrated that the reaction did not proceed without Lewis acid catalyst Co(OTf)₂ and a single ligand **L10** was unable to catalyze the reaction without Co(OTf)₂. Utilizing Co(OTf)₂ alone resulted in a 99% yield of the desired racemic **3ab**, highlighting the essential role of the Lewis acid catalyst. Corresponding information is shown in ESI Table S1, entries 24-26.

12: The identification of key intermediates is an important part of describing the catalytic mechanism. The key intermediates in the reaction process are characterized and experimentally confirmed. DFT calculations can help to elucidate the catalytic mechanism by simulating the reaction process and obtaining its reactive intermediates, transition states, and so on. In the manuscript, a diagram of the catalytic mechanism should be presented.

Response 12: Thank you for your suggestion. Deng recently conducted DFT mechanistic studies on the Lewis-catalyzed racemic (3+3) cycloadditions involving BCBs and nitrones (ref 29). Additionally, Xie and Guo's group also performed DFT mechanistic studies on the Co(OTf)₂-PyIPI **L12**-catalyzed asymmetric dearomative [3+2] annulation of quinolines and donor-acceptor aminocyclopropanes (ref 67). Building upon their comprehensive mechanistic investigations, we utilized density functional theory (DFT) calculations to elucidate the enantiocontrol mechanism in our asymmetric (3+3) cycloadditions of BCBs and nitrones (Fig 6).

13: The formation of complexes between BCB and Lewis acid catalysts helps to enhance enantioselectivity. How is the intermediate complex/transition state determined? The intermediate structure can be well speculated using DFT computational simulations.

Response 13: Thank you for your suggestion. As per your request, we utilized density functional theory (DFT) calculations to elucidate the enantiocontrol mechanism in our asymmetric (3+3) cycloadditions of BCBs and nitrones (Fig 6).

Response to Referee (3)'s Comments -- changes highlighted in blue.

Comments:

In this manuscript, Feng and co-workers described the first example of (highly) enantioselective formal [4+2] cycloaddition of BCBs with nitrones using a chiral Lewis acid catalyst. Addressing the inherent difficulties of such a reaction required a meticulous screening of conditions. Key to success was the combination of a Co(II) salt with an unusual tridentate nitrogen ligand of the PyIPI family. Substrates incorporating imidazole or pyrazole substituents were used in order to ensure the coordination of the latter to the catalyst. A monomeric transition state was proposed based on non-linear effect studies. Consistently, the reaction delivered enantio aza-oxa-bicycloheptanes, which are interesting as saturated analogs of disubstituted pyridines. These products could be converted into a series of valuable derivatives, including a densely functionalized cyclobutanol. The scope of the reaction was broad: a diversity of substituents on both the BCBs (bridgehead position) and the nitrones (N and C positions) were well tolerated. Enantiomeric excesses as high as 99% and good to excellent yields were obtained in most cases.

Overall, Feng and co-workers present some elegant and solid chemistry. Assessing the novelty delivered by their manuscript was initially less trivial, because of the analogies with the communication published by Deng and co-workers in *Angewandte Chemie* earlier this year (ACIE 2024, 63, e202318476). In fact, the latter described a very similar annulative reaction involving BCBs and nitrones. Upon accurate reflection, I nonetheless got to the conclusion this occurrence does not invalidate the originality or the significance of Feng's work, which are mainly residing in the asymmetric character of the transformation. Enantio-induction in annulations of BCBs is extremely challenging to achieve, especially under Lewis acid catalysis. Providing a well-defined chiral environment through the coordination of the catalyst is though, so that background reactions can easily dominate. So far, the only reports depicting asymmetric annulations of BCBs have been the ones by Bach and, respectively, Jiang. These valuable achievements, which relied on photocatalysis, were however limited to formal [2+2] cycloadditions and subjected to structural constraints of the reacting species. The robust and general method established by Feng and co-workers indicates that a broader use of bicyclobutanes in enantioselective annulation is feasible, which is highly desirable when investigating the resulting cycloadducts for drug discovery purposes.

Based on these considerations, I therefore strongly encourage the publication of this manuscript by Feng and co-workers in *Nature Communications*. The authors are however recommended to implement the following revisions:

Response: We appreciate the time and effort you dedicated to providing feedback on our manuscript. Thank you for your positive comments and valuable suggestions to improve the quality of our manuscript. As the principal investigator (PI) of a newly established research group, I am delighted that you, a prominent figure in the field, have acknowledged our work, and your encouragement is highly valued.

1: It has now become common to read of $[n\pi+2\sigma]$ in reports describing annulations with bicyclobutanes. Calling the electrons coming from the latter as 2σ is partially erroneous, because the bond across BCBs has a ca. 95% π character, as computed by Newton and Schulman already in 1972 (JACS 1972, 94, 767). The authors should therefore change $[4\pi+2\sigma]$ into $[4+2]$ everywhere in their manuscript to provide a less ambiguous definition of the process.

Response 1: Thank you for your valuable suggestion. To avoid ambiguity in defining

cycloadditions, we prefer using the conventional nomenclature for these reactions. The notation [4+2] in square brackets indicates the electron count of the two reacting fragments, while (3+3) in round brackets denotes the atom count. Throughout the manuscript, we have replaced [4 π +2 σ] with (3+3).

2: When examining the scope, the variations of the BCB substrates are limited to changing the aryl substituent at the bridgehead position of the bicyclic scaffolds. If not already done, the authors should also tested H- and/or alkyl substituted BCBs and comment about the corresponding results.

Response 2: Thank you for your suggestion. The current reaction did not apply to methyl-substituted BCB substrate **1r** (up to 52% ee) and mono-substituted BCB substrate **1t**. The ester-substituted BCB **1u** was used in the (3+3) cycloaddition but did not produce any cycloadduct, indicating that the chelation of bidentate group to Lewis acid catalyst is essential. Corresponding information is shown in Scheme S5 in ESI.

3: The manuscript is well written. The abstract, the introduction and the conclusions are clear, and offer an adequate context to justify the importance of the reaction. The Supporting Information, on the other side, requires some adjustments. The authors should:

-- Give synthetic protocols for the preparation of all the tested chiral ligands that are non-commercially available, including the related characterization data.

-- Check and correct the significant digits throughout the document, as they are not consistent.

-- Provide the melting points of all the previously unreported compounds that are solid.

-- Concerning the linear effect experiment, error bars should be displayed to prove the statistical significance of the shown results.

Response 3: Thank you for your valuable suggestion. As per your request, the origins of the ligands discussed in the article are detailed within the ESI. Synthetic protocols for preparing ligands **L9-12** and **L21** are provided. Significant digits have been checked and corrected throughout the document. Melting points of the solid products are also documented in the ESI.

Error bars have been included in the linear effect experiment.

4: The authors provided an exhaustive list of citations. References 39 and 40 appear however poorly fitting in the context in which they are given. Rephrasing lines 55-56 in page 4 may be appropriate, by mentioning that – prior to Bach and Jiang – the only asymmetric difunctionalization reactions across the BCB framework were limited to ring opening processes.

***Response 4:** Thank you for your suggestion. As per your request, the corresponding sentence has been rephrased as follows:” While chirality transfer and bio-catalysis strategies have been used to synthesize chiral BCPs and BCHs respectively, before the groundbreaking asymmetric (3+2) cycloadditions of BCBs reported by Bach and Jiang, the only asymmetric difunctionalization reactions within the BCB framework were limited to ring opening processes.”*

REVIEWERS' COMMENTS

Reviewer #2 (Remarks to the Author):

After the first round of reviews, the authors and the research team have made significant improvements to their manuscript and a further added a great deal of necessary work, which include a considerable amount of experimentation and exposition. I have no further suggestions for this work - there is no doubt that the catalysts formed by the combination of bidentate ligand BCB and chiral Lewis acids exhibit excellent catalytic properties and enantioselectivity. Also, this work will be of greater interest and appeal to readers in synthetic chemistry, medicinal chemistry and chiral chemistry. In order to disseminate this result to the community for further research, I recommend publication without further revision.

Reviewer #3 (Remarks to the Author):

Feng and co-workers described the annulation of bicyclobutanes with nitrones, achieving high levels of enantioselectivity by using a Lewis-acid Co(II) catalyst with a chiral ligand. This accomplishment is a breakthrough in this fast growing and greatly important area of organic chemistry, where most reports have so far focused on annulative reactions of BCBs occurring in a racemic manner.

Compared to the version of the manuscript that had been previously submitted, I find that the authors addressed satisfyingly the comments of my two other fellow reviewers as well as mine. In particular, the ESI have been complemented with the requested additional details and are now more exhaustive.

I can therefore confirm my recommendation that the manuscript should be published in Nature Communications.

Response to Referees

Response to Referee (2)'s Comments

Comments:

After the first round of reviews, the authors and the research team have made significant improvements to their manuscript and a further added a great deal of necessary work, which include a considerable amount of experimentation and exposition. I have no further suggestions for this work - there is no doubt that the catalysts formed by the combination of bidentate ligand BCB and chiral Lewis acids exhibit excellent catalytic properties and enantioselectivity. Also, this work will be of greater interest and appeal to readers in synthetic chemistry, medicinal chemistry and chiral chemistry. In order to disseminate this result to the community for further research, I recommend publication without further revision.

Response: We appreciate the time and effort you dedicated to providing feedback on our manuscript. Thank you for your positive comments and valuable suggestions to improve the quality of our manuscript. As the principal investigator (PI) of a newly established research group, I am delighted that you, a prominent figure in the field, have acknowledged our work, and your encouragement is highly valued.

Response to Referee (3)'s Comments

Comments:

Feng and co-workers described the annulation of bicyclobutanes with nitrones, achieving high levels of enantioselectivity by using a Lewis-acid Co(II) catalyst with a chiral ligand. This accomplishment is a breakthrough in this fast growing and greatly important area of organic chemistry, where most reports have so far focused on annulative reactions of BCBs occurring in a racemic manner.

Compared to the version of the manuscript that had been previously submitted, I find that the authors addressed satisfyingly the comments of my two other fellow reviewers as well as mine. In particular, the ESI have been complemented with the requested additional details and are now more exhaustive.

I can therefore confirm my recommendation that the manuscript should be published in Nature Communications.

Response: We appreciate the time and effort you dedicated to providing feedback on our manuscript. Thank you for your positive comments and valuable suggestions to improve the quality of our manuscript. As the principal investigator (PI) of a newly established research group, I am delighted that you, a prominent figure in the field, have acknowledged our work, and your encouragement is highly valued.